Construct validity and internal consistency of the Breast Inflammatory Symptom Severity Index in lactating mothers with inflammatory breast conditions

Heron Emma emma.duff@postgrad.curtin.edu.au 1
McArdle Adelle 2
Karim Md Nazmul 3
Cooper Melinda 4
Geddes Donna 5
McKenna Leanda 1
1 School of Allied Health, Curtin University , Bentley , Western Australia , Australia
2 Monash Rural Health, Monash University , Churchill , Victoria , Australia
3 School of Public Health and Preventative Medicine, Monash University , Melbourne , Victoria , Australia
4 MMC Physiotherapy , Kyneton , Victoria , Australia
5 School of Molecular Sciences, University of Western Australia , Crawley , Western Australia , Australia
Prazeres Filipe
Electronic publication date: 2021 Nov 16
Publication date: 2021
Volume: 9
Electronic Location ID: e12439
Received 2021 Aug 4; Accepted 2021 Oct 15
Copyright: ©2021 Heron et al.
Copyright year: 2021
Copyright holder: Heron et al.
License: This is an open access article distributed under the terms of the Creative Commons Attribution License, which permits unrestricted use, distribution, reproduction and adaptation in any medium and for any purpose provided that it is properly attributed. For attribution, the original author(s), title, publication source (PeerJ) and either DOI or URL of the article must be cited.
License URL: https://creativecommons.org/licenses/by/4.0/

Keywords: Breastfeeding, Mastitis, Lactation, Mothers, Psychometrics, Patient reported outcome measure

Funding: Australian Government Research Training Program Scholarship Emma Heron undertook this work as part of the completion of a PhD and was supported by an Australian Government Research Training Program Scholarship (https://www.dese.gov.au/research-block-grants/research-training-program). The funders had no role in study design, data collection and analysis, decision to publish, or preparation of the manuscript.

==============================
Background

Inflammatory Conditions of the Lactating Breast (ICLB) affect more than one in five lactating mothers, yet no fully validated outcome measures exist to aid clinicians in their patient-centred care of women with ICLB. The Breast Inflammatory Symptom Severity Index (BISSI) is an ICLB-specific clinician administered patient-reported outcome measure, currently used by Australian clinicians, who treat mothers with ICLB. To date the BISSI has undergone partial psychometric development. This study, therefore, aimed to undertake the next stage of psychometric development by determining the construct validity and internal consistency of the BISSI.

Methods

A retrospective audit was conducted on patient records of 160 mothers who were treated for ICLB, at a private physiotherapy practice in Melbourne, Australia. An electronic data capture tool was used to collate BISSI scores and associated ICLB assessment variables. Construct validity was determined through factor analysis and discriminant performance. Reliability was determined by assessing measures of internal consistency.

Results

Factor analysis established that BISSI items (n = 10) loaded on to four factors, Wellness, Pain, Physical Characteristics of Affected Area (PCAA), and Inflammation, which together, explained 71.2% of variance. The remaining item (‘Wellness/sickness unspecified’) did not load. Wellness, Pain, PCAA and Inflammation factors individually and collectively displayed the ability to discriminate symptom severity, as scores were significantly higher in mothers with high symptom severity (assessed via AUC close to or >0.7 and P value <0.005 for each factor). The BISSI demonstrated internal consistency with an overall Cronbach alpha of 0.742.

Conclusions

The BISSI has adequate construct validity, demonstrating behaviour consistent with theoretical constructs of inflammation severity, via its dimensionality and ability to discriminate symptom severity. The BISSI also has adequate internal consistency demonstrating reliability. Therefore, clinicians can have confidence that the BISSI is valid, the individual item scores are correlated, and the concepts are consistently measured.

Introduction

Inflammatory conditions of the lactating breast (ICLB) include non-physiological engorgement, blocked ducts, mastitis, and breast abscess (Heron et al., 2020). These conditions are all characterised by a combination of local, and/or systemic signs of inflammation such as breast pain, redness, swelling, heat, and fever and flu-like symptoms (LactaResearch Group, 2018). Clinical presentations of ICLB vary greatly, from a few local breast symptoms to rapid onset of acute physical illness, that may substantially interfere with a mother’s physical and emotional daily functioning (Amir & Lumley, 2006). The exact aetiology and role of bacterial pathogens in the clinical manifestations of ICLB is unclear and has been debated over the last three decades (Collado et al., 2009; Ingman, Glynn & Hutchinson, 2014; Kvist et al., 2008). Scientific evidence now suggests that bacterial species may not be the primary causative agents, and rather, ICLB may be the result of a transient alteration of the mother’s milk microbiome, along with genetic and environmental factors (Delgado et al., 2008; Hunt et al., 2011; Ingman, Glynn & Hutchinson, 2014; Jahanfar, Ng & Teng, 2013; LaTuga, Stuebe & Seed, 2014). A recent systematic review (Wilson, Woodd & Benova, 2020) found that approximately one in four lactating mothers are affected by mastitis during the first six months postpartum. Consequently, many mothers present to clinicians in the early postpartum period, seeking relief for their often debilitating symptoms of breast inflammation.

In Australia, mothers with ICLB commonly seek treatment from general practitioners, lactation consultants, general and women’s health physiotherapists, and hospital emergency departments. Each profession provides different treatments for ICLB, although their roles may overlap. Common treatments for ICLB provided by Australian physiotherapists include therapeutic ultrasound, education and advice, massage, Tubigrip®, kinesiology tape and low-level laser therapy (Diepeveen et al., 2019). The absence of a fully validated clinical measure for ICLB limits clinicians’ capacity to appropriately assess mothers, monitor their treatment response, and follow their clinical progress over the course of their condition (Kyte et al., 2014). Current clinical practice is comprised of subjective assessment of symptoms, which forms the basis for diagnostic and treatment decision making (Amir, Trupin & Kvist, 2014). Therefore, a psychometrically robust tool to measure the most important clinical presentations of ICLB is required. Accurate assessment of the cardinal signs and symptoms of inflammation (pain, redness, swelling, heat and loss of function) (Scott et al., 2004) and the changes in these inflammatory symptoms that may be attributed to treatment will aid in the care for women with ICLB. Validated outcome measures for use in ICLB interventional clinical trials, are also required to help develop high-level evidence for ICLB treatments. Such a tool would also be useful to improve and provide impetus for future international research on ICLB.

The Breast Inflammatory Symptom Severity Index (BISSI) is an ICLB-specific Clinician Administered Patient Reported Outcome Measure (CAPROM) (Cooper, Lowe & McArdle, 2020). It is currently in clinical use by clinicians who have attended the Australian ‘Lactation for Health Professionals’ course (Inform Physiotherapy, 2021). The BISSI was originally developed to provide clinicians and mothers with a simple, quick, immediate, and prognostic measure of symptom severity, capturing all the inflammatory symptoms. It is administered at the time of consultation and uses a Numerical Rating Scale (NRS) to measure pain, systemic symptoms, and functional impact; and a 5-point scale to measure breast hardness/tightness (swelling), breast temperature, redness, and size of the affected area (see Table 1). An accompanying clinician script was developed to preserve face validity and utility (Cooper, Lowe & McArdle, 2020). Ease of utility was deemed particularly important, since mothers with ICLB can be acutely unwell and clinicians administering the tool have varied experience with ICLB (Cooper, Lowe & McArdle, 2020). The BISSI is the only ICLB PROM to have undergone partial psychometric development. While its face and content validity have been recently established (Cooper, Lowe & McArdle, 2020), further psychometric evaluation is required.

Table 1 Breast Inflammatory Symptom Severity Index (BISSI) assessment items.

BISSI items	Scale	Inflammatory symptom	
1. Pain*			
a) Awareness	11-point NRS scale	Breast pain (on awareness)	
b) Touch	11-point NRS scale	Breast pain (on touch)	
2. Wellness/Sickness#			
a) Fever	11-point NRS scale	Systemic symptoms	
b) Generalised aches & pains	11-point NRS scale	Systemic symptoms	
c) Headache	11-point NRS scale	Systemic symptoms	
d) Sickness (unspecified)	11-point NRS scale	Systemic symptoms	
3. Hardness/Tightness	5-point Likert scale	Breast swelling due to ICLB	
4. Temperature of affected area	5-point Likert scale	Breast heat due to ICLB	
5. Redness	5-point Likert scale	Breast redness due to ICLB	
6. Affected Area	5-point Likert scale	Size of affected breast area	
7. Impact	11-point NRS scale	Functional loss	
Total score:			
Awareness	Maximum of 80		
Touch	Maximum of 80		
Notes.

Note: In the BISSI there are seven questions, two of which have sub-questions*#. In this paper we consider all sub-questions as an item. Thus, there were 11 items from seven questions of which one question* had two sub-questions and another question# had four sub-questions. The remaining five were stand-alone questions.

The aim of this study was to determine the psychometric properties of the BISSI to further develop the tool for use in clinical settings and ICLB efficacy trials (Cooper, Lowe & McArdle, 2020). Specifically, an exploration of construct validity, via an examination of the dimensionality and ability to discriminate severity of symptoms was undertaken. An assessment of the reliability of the BISSI, through an examination of internal consistency, was also undertaken.

Materials & methods

A retrospective audit of patient clinical notes of mothers with ICLB was performed at a private physiotherapy practice in Melbourne, Australia. Ethical approval for this study was granted by Curtin University Human Research Ethics Committee (HRE2020-0544) which included a waiver of consent.

Participants/sample

Clinical appointment notes from 160 lactating mothers who presented to the private physiotherapy practice between 12 July 2017 and 15 September 2020 with an ICLB, were examined (Fig. 1). Clinical notes from an appointment were considered eligible if the mother had presented with ICLB, was over 18 years of age and had an accompanying completed record of their BISSI scores at their initial appointment. The initial appointment was defined as the first appointment or contact with the clinician for care of a defined episode of ICLB. An ICLB episode was the period from initial onset of ICLB symptoms until complete resolution. Data for one ICLB episode per mother was collected. The practice management software (Cliniko, Melbourne, Australia) (Cliniko, 2010), was used to identify and scrutinise eligible clinical notes.

Figure 1 Eligibility.

A search of all ICLB appointment types (initial, review and extended) was conducted in Cliniko from 15 September 2020 until 160 eligible case notes were identified. The auditor (EH), an experienced Women’s Health physiotherapist registered with Australian Health Practitioner Regulation Agency, used the Cliniko appointment diary to search on a day-by-day basis, with the ‘hide names’ function activated to maintain anonymity of non-ICLB patients. Appointment types were colour-coded in Cliniko, hence ICLB appointments could be identified. An episode was identified by locating an initial appointment or clear documentation of a new ICLB episode within the clinical record. Where multiple episodes of care occurred, only the data from the most recent ICLB episode was used.

The BISSI has 11 assessment items (see Table 1), producing eight individual item scores. The statistical analysis required 10 to 20 scores per item to produce stable factor analysis solutions with reduced sampling error (Maccallum et al., 1999; Thompson, 2004). Therefore, based on the sample size recommendations in factor analysis, 160 complete BISSI scores or eligible clinical notes were examined, derived from eight scores multiplied by 20.

Procedure

Data collection was performed in October 2020. Research Electronic Data Capture, a secure, web-based software platform designed to support data capture for research studies, hosted at Curtin University, was used to collect, and manage the data (Harris et al., 2019; Harris et al., 2009; REDCap 2004). Forced response and validation rules were used, to minimise data omission errors. Three data collection variable domains were created. The first domain comprised demographic data of maternal date of birth, maternal parity, breastfeeding infant’s age, mode of delivery, single/multiple birth, month of presentation and socioeconomic status, including postcode, private health insurance and maternal occupation. With respect to mother’s occupation, an Occupational Socioeconomic Status Scale, modified from Marks et al. (2000), was used to classify occupational data (see Table S1). The scale consists of six groups, with four hierarchical occupation levels (group 1 to 4 respectively) based on required skill level and skill specialisation. Group 5 represented those not currently in paid work, and group 6 represented occupation unreported. Three members of the research team (EH, AM, LM) independently used the scale to classify 66 different occupations named by the mothers. Where discrepancies (29) occurred, majority consensus was used to assign a score. For the postcode socioeconomic status measure, the Australian Bureau of Statistics, Socio-Economic Indexes for Areas 2016, Index of Relative Socio-Economic Advantage and Disadvantage was used. Within the index, the postal area decile ranking within Australia was used, which ranks postal area codes from lowest to highest advantage, with a decile number of 1 representing the lowest 10% of postal areas (most disadvantaged) up to a decile number of 10, the highest 10% of postal areas (most advantaged).

The second domain of ICLB characteristics comprised the affected breast (right or left) and quadrant, number and type of local and systemic symptom(s), including symptom onset, and antibiotic use. The third domain of clinician assessment comprised BISSI scores and clinician breast and nipple observation (including number and type of local breast inflammatory symptoms).

Analysis

Pre-analysis data screening demonstrated no violations to the assumptions of linearity, normality, multicollinearity, and homoscedasticity. All individual item scales were standardised for maximal scale length for the analysis, given the heterogeneity across the BISSI items (see Table 1). Standardisation occurred by converting all Likert scales into the 11-point NRS scale (0–10), similar to what is used to measure pain (Karcioglu et al., 2018), creating scale homogeneity. This ensured individual items contributed equal weight to the total score.

Descriptive statistics were generated for maternal demographics and characteristics, and BISSI items. Exploratory Factor Analysis (EFA) using Principal Component Analysis (PCA) was used to identify underlying factors within the BISSI (Field, 2009; Streiner, 2015). Confirmatory Factor Analysis (CFA) cross-validated the factor structure derived from EFA, to establish the dimensionality of the BISSI. Items of the BISSI that loaded onto the same factors were combined as a ‘factor’ and factor-specific validity and reliability was assessed subsequently. Individual items were screened to identify those with poor factor loading (<0.3).

Discriminant validity was explored by assessing the BISSI’s ability to discriminate between mothers with high and low symptom severity. Mothers with mastitis were considered likely to have the highest scores or severity. Other conditions within the suite of ICLB are not considered to be as severe (Betzold, 2007), providing a suitable differentiation against mastitis for discriminant analysis. A diagnosis of mastitis (Amir & Academy of Breastfeeding Medicine Protocol Committee, 2014; Amir, Trupin & Kvist, 2014), as being “…at least 2 breast signs/symptoms (pain, lump/hardness, redness) and fever or at least 1 systemic symptom (lethargy, aching, headache, nausea and so on)”, determined mothers of high symptom severity.

Reliability (measured using internal consistency) of the BISSI was assessed using Cronbach’s α coefficient (Field, 2009). A Cronbach alpha value of 0.70 or above was considered an acceptable level of internal consistency (Taber, 2018). The contribution of each item and factor on the BISSI was assessed by generating the item-total and factor-total correlation and by computing Cronbach’s alpha excluding that item or factor, respectively.

All statistical analyses were performed using statistical software R-3.6.0 (The R Project for Statistical Computing) (R Core Team, 2017), and STATA/IC release 15 (StataCorp, 2017) where appropriate.

Results

Demographics

A total of 290 ICLB clinical notes from 197 mothers were identified initially and examined for eligibility (Fig. 1). There were 21 mothers with ineligible clinical notes, due to no or incomplete initial appointment BISSI scores, and 16 mothers who had more than one identified ICLB episode during the collection period. For these 16 mothers, their most recent ICLB episode case notes were included in this study and prior episodes were excluded.

Demographics for the 160 included mothers are presented in Table 2. In general, mothers were aged in their thirties and their breastfeeding infants ranged from 4 days to 21 months old. Most mothers had a high socioeconomic status, as they lived in the most advantaged postal areas, had private health insurance, and were employed in professional occupations. In unilateral ICLB presentations, the right breast was most affected, and the most common location on the right breast was the inferior lateral quadrant (see Fig. 2). Nearly half the mothers matched the criteria (Amir, Trupin & Kvist, 2014) for diagnosis of mastitis during their current ICLB episode.

Table 2 Mothers’ demographics (N = 160).

Demographic variable	Median (Q1, Q3) or n (%)	
Maternal agea (years)	35 (31, 37)	
Range	25–42	
Maternal parityb		
Primiparity	85 (53.1)	
Multiparityc	68 (42.5)	
Two children	58 (36.3)	
Three children	8 (5)	
Singleton birth	160 (100)	
Mode of deliveryd		
Vaginal	90 (56.3)	
Caesarean	37 (23.1)	
Socioeconomic status		
Postal area indexe		
1f	0 (0)	
2	5 (3.1)	
3	4 (2.5)	
4	11 (6.9)	
5	2 (1.3)	
6	2 (1.3)	
7	26 (16.3)	
8	6 (3.8)	
9	37 (23.1)	
10g	59 (36.9)	
Private health insurance		
Yes	111 (69.4)	
Unknown	49 (30.6)	
Occupation groupingh		
1	65 (40.6)	
2	60 (37.5)	
3	11 (6.9)	
4	0 (0)	
5	7 (4.4)	
6	17 (10.6)	
Month of presentation		
Jan	12 (7.5)	
Feb	10 (6.3)	
Mar	20 (12.5)	
Apr	8 (5)	
May	14 (8.8)	
Jun	11 (6.9)	
Jul	14 (8.8)	
Aug	22 (13.8)	
Sept	18 (11.3)	
Oct	10 (6.3)	
Nov	14 (8.8)	
Dec	7 (4.4)	
Affected breast		
Left	67 (41.9)	
Right	73 (45.6)	
Bilateral	20 (12.5)	
Infant age (weeks)	9.5 (4.0, 21.7)	
Range	0.57–91.25	
Symptom onset (days ago)	2 (1, 3)	
Episode systemic symptomsi		
Yes	84 (52.5)	
No	21 (13.1)	
Not recorded	55 (34.4)	
Mastitis		
Episode		
Yes	79 (49.4)	
No	44 (27.5)	
Unable to determine	37 (23.1)	
Initial appointment		
Yes	54 (33.8)	
No	56 (35.0)	
Unable to determine	50 (31.3)	
Antibiotic usej		
Yes	72 (45)	
No	48 (30)	
Not recorded	40 (25)	
Notes.

Q= Quartile.

a n = 2 maternal date of births not reported.

b n = 7 not reported.

c n = 2 not reported.

d n = 33 not reported.

e n = 8 not reported.

f Most disadvantaged.

g Most advantaged.

h Occupational socioeconomic status scale–modified version of Marks et al. (2000): Group 1–Senior management and qualified professionals; Group 2–Other managers and associate professionals; Group 3–Trades people and skilled staff; Group 4–Assistants and labourers; Group 5–Not currently in paid work; Group 6–Not reported (see Table S1).

i Any indication in the initial clinic notes (excluding BISSI scores), that the mother had systemic symptoms for this ICLB episode.

j At initial appointment, for this ICLB episode.

Figure 2 Affected breast quadrants.

Construct validity

Factor analysis

Preliminary exploratory factor analysis (EFA) revealed the item ‘Wellness/sickness unspecified’ had a factor loading of less than 0.3. Confirmatory factor analysis extracted four distinct factors, Pain (incorporating items Pain Awareness and Pain Touch), Wellness (incorporating items Fever, Ache, Headache), Physical Characteristics of Affected Area (PCAA) (incorporating items Hardness, Area, Impact) and Inflammation (incorporating items Redness and Temperature) (Fig. 3, see correlations in Table 3). The four factors together explained 71.2% of the variation in the score (Table 3).

Figure 3 Bar chart illustrating factor loading.

Table 3 Factor loading of BISSI for the extracted factors through principal component analysis (n = 160).

Item
number	Factor	Item description	Extracted factors (correlations)	
			1	2	3	4	
1	Wellness	Headache	0.727				
2	Wellness	Ache	0.806				
3	Wellness	Fever	0.838				
4	Pain	Touch		0.856			
5	Pain	Awareness		0.892			
6	PCAAa	Impact			0.675		
7	PCAAa	Affected area			0.720		
8	PCAAa	Hardness			0.765		
9	Inflammation	Temperature				0.833	
10	Inflammation	Redness				0.861	
		Eigenvalues of Factors	3.255	1.554	1.208	1.103	
		% variance explained by factors	32.552	15.54	12.082	11.028	
Notes.

Extraction Method: Principal Component. Rotation Method: Varimax with Kaiser Normalization.

a PCAA, Physical characteristics of affected area.

Discriminant validity

The BISSI and all its factor scores were found to be significantly higher in mothers with high symptom severity (Table 4). The Area under the Curve (AUC) analysis (factors of Wellness, Pain, Inflammation, PCAA and BISSI), indicated the BISSI can correctly identify high symptom severity in close to or greater than 70% of the participants (Table 5). The Receiver Operating Characteristic (ROC) analysis (Fig. 4) indicates a good level of discriminatory accuracy for the BISSI and all factors.

Table 4 Differences in the BISSI item scores and total score between mothers with high or low symptom severity.

Factors	Low	High	P Value	
	Mean	SD	Mean	SD		
Wellness	1.7	4.7	7.4	7.0	0.000002	
Pain	5.3	3.7	8.6	4.7	0.000069	
PCAAa	18.1	5.0	20.7	4.5	0.006078	
Inflammation	7.8	2.8	12.7	4.1	<0.000001	
Total score	32.9	10.3	49.4	13.2	<0.000001	
Notes.

a PCAA, Physical characteristics of affected area.

Table 5 Area under the Curve (AUC) analysis of mothers with high symptom severity.

Factors	AUC	P Value	95% CI	
			Low	High	
Wellness	.770	.000001	0.679449	0.860895	
Pain	.704	.000222	0.607184	0.801216	
PCAAa	.648	.007592	0.543919	0.751386	
Inflammation	.832	.000001	0.756933	0.906758	
BISSI	.857	.000001	0.784782	0.929503	
Notes.

a PCAA, Physical characteristics of affected area.

Figure 4 ROC analysis of high symptom severity.

Internal consistency

The BISSI showed high internal consistency (Table 6). The overall Cronbach’s α coefficient for the BISSI was above 0.7 indicating the BISSI was reliable and repeatable (Table 6). Item-total correlation across items ranged from 0.25 to 0.56 (Table 7). All the items either caused a decrease or no change in the overall BISSI Cronbach’s α values upon removal from the BISSI, except for the items Fever, Hardness, and Redness (Table 7).

Table 6 Internal consistency analysis of BISSI and the proposed factors.

Factor	Item numbers	Descriptive statistics Mean (sd)	Cronbach’s alpha	Factor total correlation coefficient	
Total score	1 –10	40.1 (13.3)	0.742	–	
Wellness	1 –3	3.7 (5.9)	0.754	0.743	
Pain	4, 5	6.6 (4.3)	0.784	0.675	
PCAAa	6 –8	19.1 (4.9)	0.586	0.681	
Inflammation	9, 10	10.1 (4.1)	0.720	0.611	
Notes.

a PCAA, Physical characteristics of affected area.

Table 7 Internal consistency analysis of BISSI items.

Item
Number	Item description	Item
Mean (SD)	BISSI Mean if Item Deleted	Corrected Item-Total Correlation	Cronbach’s Alpha if Item Deleted	
1	Fever	0.8 (2.1)	38.8	0.35	0.753	
2	Aches	1.4 (1.4)	38.1	0.56	0.725	
3	Headache	1.5 (1.5)	38.0	0.50	0.732	
4	Pain Awareness	2.1 (2.1)	37.5	0.44	0.742	
5	Pain Touch	4.6 (2.6)	35.0	0.52	0.730	
6	Hardness	4.9 (2.3)	34.6	0.37	0.751	
7	Area	6.3 (1.8)	33.3	0.42	0.747	
8	Impact	6.6 (2.5)	33.0	0.44	0.742	
9	Redness	6.2 (2.3)	33.4	0.25	0.768	
10	Temperature	5.1 (2.3)	34.4	0.42	0.744	

All factors showed internal consistency with individual factor Cronbach’s α values for Wellness, Pain, Inflammation and BISSI (total score) factors above the acceptable 0.7 Cronbach’s α value for internal consistency (Table 6). The Cronbach’s α value for the factor of PCAA (incorporating items Hardness, Area, and Impact) was just under 0.7 (Table 6).

Discussion

This retrospective audit of clinical notes from mothers with ICLB, provides evidence supporting the validity and reliability of the BISSI. Discriminant validity and internal consistency were established, demonstrating the ability of the BISSI to discriminate and reliably measure symptom severity in mothers with ICLB. Overall, the clinical tool performed well psychometrically.

The factor analysis revealed a four-factor structure underlying 10 of the 11 items on the BISSI, with one item, ‘Wellness/Sickness unspecified’, not contributing to the factor structure. This item originally allowed mothers to rate their degree of systemic ‘wellness or sickness’ without specifying symptoms, if they did not have the already specified symptoms of fever, generalised aches and pains, or headache. Measuring a mother’s ‘worst’ sickness using the ‘Wellness/Sickness unspecified’ item was found to be imprecise when measuring ICLB symptoms, presumably because many reasons can influence a mother’s state of wellness. Therefore, it is proposed that the item ‘Wellness/Sickness unspecified’ be removed from the BISSI.

The three specific systemic symptoms of fever, generalised aches and pains and headache, strongly loaded on to the factor ‘Wellness’. These findings align with mothers’ reports of ICLB, which commonly include the presence of flu-like symptoms, such as fever, aches and pains, and/or headache (Cooper, Lowe & McArdle, 2020; Heron et al., 2020; Kvist, 2006). The BISSI asks mothers to select their worst symptom of the three and rate the severity of this symptom only. However, the mother’s selected symptom can differ on subsequent BISSI responses. Therefore, it is proposed that these three symptoms of fever, aches and pains, and headache, are all included within the BISSI item wording, to give one all-encompassing severity rating for these ICLB systemic symptoms.

The BISSI items of Impact, Affected Area and Hardness/Tightness loaded on to the factor ‘PCAA’, indicating a correlation between the severity, size of affected area, swelling or hardness/tightness, and impact/interference on the mother’s everyday life. This loading is consistent with inflammation theory which includes the fifth cardinal sign and symptom, loss of function (Scott et al., 2004). The BISSI is therefore measuring the important cardinal signs and symptoms of inflammation, and no change is proposed to these items on the BISSI as a result of these findings.

As both pain items (awareness and touch) loaded strongly on to the factor ‘Pain’, we propose incorporating both pain items in the final index score of the BISSI. It may be important to maintain the separate distinction of pain on ‘awareness’ versus ‘touch’ by incorporating both in the final score, as pain on awareness may be a manifestation of peripheral sensitisation. Additionally, a high score against ‘pain awareness’ may indicate the condition is developing central modulation (Baron, Hans & Dickenson, 2013). This change is supported by prior findings, whereby pain was rated as the most important symptom in ICLB, and the symptom perceived to change the fastest in response to treatment (Cooper, Lowe & McArdle, 2020). Importantly, all proposed changes align with previous research, wherein mothers indicated that the tool must capture their range and severity of symptoms and concerns, while also being concise and accurate (Cooper, Lowe & McArdle, 2020). The BISSI may help the clinician to determine if ICLB severity is worsening or abating and thus whether treatments are effective. The final proposed change to the BISSI includes scoring all items similarly using an NRS. This was a change implemented for statistical analysis conducted in this study, to ensure equal weighting of individual items in the BISSI.

The BISSI demonstrated good construct validity and can discriminate based on symptom severity. Clinically, this shows the BISSI can distinguish between mothers with high and low ICLB symptom severity. This means clinicians can have greater confidence that the BISSI measures the theoretical construct of symptom severity. The BISSI also demonstrated internal consistency, indicating it is a reliable measure and clinicians can have confidence that the items reliably measure a similar construct.

This is the second study to have measured psychometric properties of the BISSI, providing another important step towards the development of a psychometrically robust ICLB-specific CAPROM for clinicians and researchers. Further psychometric testing of the BISSI is required to prospectively assess the tool’s responsiveness to change over time and determine its’ convergent and criterion validity. Further measurement and analysis of the BISSI should aim to determine the weighting of individual items along with the contribution of each item to the total severity index score. Future analysis of the BISSI would help ensure standardised clinical data is recorded, alongside accepted criterion outcome measures.

While a strength of this study is its sample size, it is limited by its retrospective cross-sectional design and the potential for selection bias. Mothers were selected from one private physiotherapy practice, and most were of high socioeconomic status, and may not be representative of all mothers with ICLB. Additionally, different clinicians were often involved in the mother’s care, as ICLB requires scheduling appointments as soon as possible with the clinician available at the time. This may have contributed to inconsistencies in documentation and outcome measurement assessment, potentially imposing additional limitations. Further, the extent of information shared by the patient and documented by the clinician may have been limited, to prioritise safe, timely treatment formulation and delivery. Thus, if a symptom was not documented, it could not be assumed it was not present. Yet, another strength of this study is its contribution to producing an internally valid and discriminatory CAPROM, which will benefit further prospective studies and enhance clinician assessment and treatment of ICLB. There is currently no gold standard outcome measure for ICLB, which has impeded assessment of criterion validity.

Conclusions

This study provides evidence for the construct validity of the BISSI by establishing its ability to discriminate the severity of inflammatory breast symptoms. Its internal consistency reliability was also established. Changes to the tool are proposed to provide an updated outcome measure for clinical use and prepare the BISSI for further psychometric evaluation in future studies. Clinicians should be trained to use the BISSI, to ensure consistent and accurate utility of this ICLB-specific CAPROM.

Supplemental Information

Supplemental Information 1 Occupational socioeconomic status scale –modified version of Marks (2000)

Click here for additional data file.

Supplemental Information 2 Deidentified raw data set

Identifying data such as maternal date of birth has been removed. The variables collected during the audit from the patient records of the 160 mothers treated for ICLB were used for the statistical analysis.

Click here for additional data file.

We wish to acknowledge the private physiotherapy practice involved in the audit, and thank the practice’s director, Hillary Schwantzer, for enabling this research.

Additional Information and Declarations

Competing Interests

Author Contributions

Human Ethics

Data Availability

The authors declare there are no competing interests.

Emma Heron conceived and designed the experiments, performed the experiments, analyzed the data, prepared figures and/or tables, authored or reviewed drafts of the paper, and approved the final draft.

Adelle McArdle conceived and designed the experiments, analyzed the data, authored or reviewed drafts of the paper, and approved the final draft.

Md Nazmul Karim and Leanda McKenna conceived and designed the experiments, analyzed the data, prepared figures and/or tables, authored or reviewed drafts of the paper, and approved the final draft.

Melinda Cooper and Donna Geddes conceived and designed the experiments, authored or reviewed drafts of the paper, and approved the final draft.

The following information was supplied relating to ethical approvals (i.e., approving body and any reference numbers):

Curtin University Human Research Ethics Committee

The following information was supplied regarding data availability:

The deidentified raw data is available in the Supplementary File.

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
