# Peer review of "Construct validity and internal consistency of the Breast Inflammatory Symptom Severity Index in lactating mothers with inflammatory breast conditions"

_PeerJ, doi:10.7717/peerj.12439_

## Round 0.1 · original submission · Major Revisions

The decision is to revise.

·

Basic reporting

Thank you for the opportunity to read this important paper on the development of an instrument to follow the progress of inflammatory symptoms of the breast during lactation. The paper is well written and the language is clear and professional. Whilst I acknowledge that this is a methodological paper (and possibly part of a future doctoral thesis) I feel that the information contained in the background is insufficient for this stand- alone paper to be properly understood by the reader. I have included some comments and suggestions regarding this below.
Introduction
The research that is cited in the background is sparse and requires a broader international perspective. The authors are surely aware that there is a history of dissent within the scientific community as to whether conditions such as mastitis are inflammatory or infectious and because this paper clearly states the measurement of inflammation as the driving force for the instrument’s development, it would be advantageous to acknowledge the fact that researchers have viewed this differently through the last 3 decades and it would enhance the understanding of the paper, if the reader was provided with references to the latest scientific approach to understanding the possible pathways of inflammatory symptoms of the breast. Here are some suggestions for literature that might be included to improve the international perspective and to give weight to the paper: (Ingman et al 2014, Martin et al 2007, Kvist et al 2008, Delgado 2008, Collado et al 2009, Hunt et al 2011, Jahanfar et al, 2013, La Tuga et al, 2014 and Human Microbiome FAQ produced by the American Academy of Microbiology).

In an international perspective, it is unusual that breastfeeding women are treated at physiotherapy clinics. The setting of this study and the background to why physiotherapists have the role of lactation consultants in Australia should be clarified, either in the background or the methods section.

On line 69 the reader is told that more than 400 clinicians are already using the instrument but without knowledge of the total number of clinicians, this statistic has no meaning.

The aim of the study is clearly stated and also that there is no existing instrument to measure ICLB. You might point out how useful such an instrument will be for future efforts to carry out international research on ICLB . This will take knowledge about ICLB forward at an increased pace. (You could include this in the conclusion or as a rationale for the research question in the introduction).

Experimental design

Materials and methods
The materials and methods are clearly presented.
On line 128, “unreported responses” seems to be an oxymoron. Does this mean that there was no answer? If so, these are not “responses”.
Lines 102-103 state that the search for eligible case notes was continued until 160 were identified. How did the authors decide on this number of case notes?
Lines 111-112 states that “the statistical analysis required a minimum of 10 to 20 scores per assessment item”. This is unclear but may just be a question of semantics; a minimum is usually given as one figure and not as a range.
Lines 151 – 153 states that ICLB has been viewed as a continuum and Betzold 2007 is cited. I contend that there is no robust empirical evidence for a continuum for ICLB. The idea of a continuum has been contested (Kvist et al., 2008). WHO (Mastitis: causes and management) suggests that breast abscess seems also to occur spontaneously, irrespective of previous mastitis, and therefore questions the idea of a continuum. This is an important point because practitioners who believe that there is a continuum will be apt to prescribe antibiotics early, to stop progress of the condition (which they would expect if there was a continuum). We are fighting a global battle against antibiotic resistance and we have a responsibility as researchers to not perpetuate unnecessary use of antibiotics.

Validity of the findings

Results
The results of the analyses appear robust and the raw data is unambiguous.
Lines 181 to 187 are headed “Analysis” and give an account of how the material was standardised. This section probably belongs to the Materials and Methods section.
Line 192: I find it odd that the items Fever, Ache and Headache are included under a factor called “Wellness” (although in the table the item is called Wellness/Sickness). These would indicate illness rather than wellness. This should be considered and changed throughout the manuscript. This is particularly visible in Table 3 where the factor is “Wellness” and the item description contains three signs of illness.

Discussion
A very nice and well-balanced discussion.

Additional comments

An important and well designed study although limited by the collection of material from only one clinic.

·

Basic reporting

No comments

Experimental design

No comments

Validity of the findings

No comments

Additional comments

The authors have done a lot of analytical work to determine the Breast Inflammatory Symptom Severity Index in lactating mothers with inflammatory breast conditions. It is wonderful. But I do not understand the purpose of this study and the practical significance of the results obtained. What are the recommendations for using this index for patients with mastitis? If it is just for making a diagnosis, then the symptoms of mastitis are so specific that it does not require the use of any severity scales. The main problem with mastitis is the rapid progression of the disease into a purulent form. At the same time, the deterioration often does not correlate with the severity of the symptoms. Therefore, the managing of mastitis (prescribing antibiotics, surgical drainage) is often determined not by the severity of symptoms, but by the time from the onset of the disease and the resolve symptoms against the background of treatment. Unfortunately, the authors did not conduct such an analysis (correlation between the index and the stage of mastitis, as well as the effectiveness of therapy).

---

## Round 0.2 · accepted · Accept

The authors have satisfactorily responded to all the questions made by the referees and made the necessary changes to the manuscript.

·

Basic reporting

No comment.

Experimental design

No comment.

Validity of the findings

No comment.

Additional comments

Congratulations on a very nice paper.